# Thinking Outside the (Gray) Box: A Context-Based Score for Assessing Value and Originality in Neural Text Generation

## Abstract

Despite the increasing use of large language models for creative tasks, their outputs often lack diversity. Common solutions, such as sampling at higher temperatures, can compromise the quality of the results. Dealing with this trade-off is still an open challenge in designing AI systems for creativity. Drawing on information theory, we propose a context-based score to quantitatively evaluate value and originality. This score incentivizes accuracy and adherence to the request while fostering divergence from the learned distribution. We show that our score can be used as a reward in a reinforcement learning framework to fine-tune large language models for maximum performance. We validate our strategy through experiments considering a variety of creative tasks, such as poetry generation and math problem solving, demonstrating that it enhances the value and originality of the generated solutions.

## 1 Introduction

Foundation models (Bommasani et al., 2021), particularly large language models (LLMs) (Gemini Team et al., 2023; Guo et al., 2025; Touvron et al., 2023), are significantly transforming creative activities. They can serve as a foundation for co-creation systems involving human and artificial authors (Lin et al., 2023), can be utilized to generate software code (Rozière et al., 2023), or even to foster scientific research (Boiko et al., 2023). However, the nature of the self-supervised learning algorithms used for the training of these models tends to make their sampling distribution as close as possible to the training data distribution (Franceschelli & Musolesi, 2024a). In addition, fine-tuning, such as that based on reinforcement learning from human feedback (RLHF) (Christiano et al., 2017), is often necessary to generate appropriate and accurate responses. However, this process tends to reduce output diversity further (Kirk et al., 2024), and linguistic creativity tends to be lower than that of humans (Lu et al., 2025). On the contrary, LLMs for creative tasks should produce more novel and surprising texts that maintain a high level of correctness and adherence to the request. One typical solution is to sample at a higher temperature to increase diversity. However, this might lead to generating less coherent text (Peeperkorn et al., 2024).

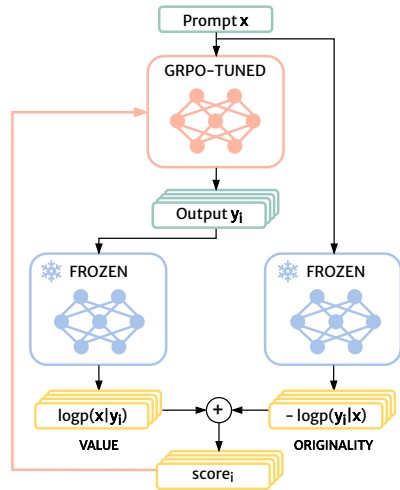

Figure 1: A summary of our method: the target model (orange) produces $G$ outputs for each prompt; a frozen reference model (blue) computes the value and originality for each output; the overall scores are used in GRPO to correct the target model (orange line).

To address the issues described above, we propose a new training approach for creative tasks based on **CoVO**, a **Co**ntext-based score for **V**alue and **O**riginality, with the goal of taking into consider-

ation both value and originality of the neurally-generated text in the optimization of LLMs. The definition of CoVO is grounded in the analysis of mutual information (MacKay, 2003) between the model's outputs and inputs, and vice versa. More specifically, we formulate a new optimization objective where, given a specific input, the desired output is derived by *simultaneously* maximizing the conditional probability of the input given the output and minimizing the conditional probability of the output given the input under the generative model. In this way, we optimize for solutions that are appropriate for the input request but also different from the outputs we would normally obtain from the model. In particular, we show that our information-theoretic score can be used as a reward in RL-based fine-tuning algorithms, guiding pre-trained models toward more diverse yet valuable solutions. Figure 1 summarizes our proposed approach.

In summary, our key contributions are the following:

- We present the theoretical foundations of our approach, deriving our context-based score for value and originality from the concept of mutual information.
- We discuss how the score can be practically computed in the case of autoregressive models, and how it can be used as a reward with Group Relative Policy Optimization (GRPO) (Shao et al., 2024), a state-of-the-art reinforcement learning algorithm for fine-tuning LLMs.
- We evaluate our GRPO-based method on mathematical problem solving, poetry generation, and the tasks included in *NoveltyBench* (Zhang et al., 2025), demonstrating that our approach can enhance both the quality and diversity of generated outputs, positioning it as a strong candidate for creativity-focused applications of current foundation models.

## 2 RELATED WORK

### 2.1 INFORMATION THEORY AND CREATIVITY

The quest to provide a mathematical and computational definition of creativity has been a significant focus in recent decades. Numerous methods have been developed to define various dimensions or attributes for evaluating the creativity of AI-generated products (see, for example, Franceschelli & Musolesi, 2024a). However, these methods are often domain-specific and typically require substantial human effort to implement and assess. In contrast, solutions based on information theory (Shannon, 1948; Cover, 1999) offer a more universally applicable approach.

Information-theoretic methods can quantify creativity by measuring the novelty and complexity of generated outputs, without the need for extensive human intervention, making them suitable for a wide range of domains. Bayesian surprise (Baldi & Itti, 2010), i.e., the divergence between a prior and a posterior belief, has been extensively used to measure different shades of originality, such as novelty (França et al., 2016; Varshney et al., 2019) and surprise (Mazzaglia et al., 2022; Schmidhuber, 2010). Nevertheless, Varshney (2019) demonstrated that there is a mathematical limit for Bayesian surprise when combined with quality measures. Surprisal (Tribus, 1961), i.e., Shannon's self-information, has also been used (Bunescu & Uduehi, 2019; Fernandez Monsalve et al., 2012); Barto et al. (2013) extensively discuss surprisal, Bayesian surprise, and novelty. Crucially, in the context of RL, surprisal has been used as a form of intrinsic motivation to encourage the agent to explore more (Achiam & Sastry, 2016). Sun et al. (2025) apply this idea to improve exploration in RLHF (Christiano et al., 2017). A similar strategy can be applied during LLM fine-tuning, either by explicitly maximizing the model's perplexity (Dai et al., 2025) or by decoupling entropy and cross-entropy from KL regularization and assigning greater weight to the former (Slocum et al., 2025). Burns (2006) proposes to use entropy for expectation and violation, plus posterior probability for explanation in the context of aesthetic experience. Additionally, mutual information has been applied to neural conversation models to improve both diversity and appropriateness (Li et al., 2016). However, all these existing approaches are not able to capture and simultaneously optimize value and originality at the same time.

### 2.2 LLMs AND CREATIVITY

Since the introduction of GPT models (Brown et al., 2020; OpenAI, 2023) and their competitors (e.g., Touvron et al., 2023), researchers have been keenly exploring the potential for LLMs to exhibit creativity and the methods to achieve this (Franceschelli & Musolesi, 2025). For example, human

creativity tests like the Alternate Uses Test have been employed to evaluate the creativity of LLMs (Stevenson et al., 2022) and to investigate methods for enhancing their performance (Goes et al., 2023; Summers-Stay et al., 2023). Porter & Machery (2024) report that non-expert poetry readers already favor AI-generated poems over human-authored ones. In contrast, Davis (2024) argues that ChatGPT's poetry is incompetent and banal. Either way, instead of being used off-the-shelf, LLMs can be fine-tuned to produce more rhyming poems (Popescu-Belis et al., 2023) or utilized in zero-shot settings to emulate the writing styles of famous authors (Sawicki et al., 2023). It has also been shown that these models can be fine-tuned via RLHF (Christiano et al., 2017) to write short poems that human evaluators find more creative (Pardinas et al., 2023). Reinforcement learning (RL) can also encourage large language models (LLMs) to produce more diverse outputs (Chung et al., 2025), and can be leveraged to optimize metrics commonly linked to creative expression (Ismayilzada et al., 2025). Finally, it is possible to leverage quality-diversity algorithms to generate more creative products; these methods can be based on human (Ding et al., 2023) or AI (Bradley et al., 2024) feedback to measure the quality of the generated outputs.

## 3 Preliminaries

### 3.1 Language Modeling

A $\boldsymbol{\theta}$-parameterized autoregressive language model is a probability distribution $p_{\boldsymbol{\theta}}(\mathbf{x})$ over a variable-length text sequence $\mathbf{x} = (x_1 \dots x_T)$, where $T$ is the sequence length and each token $x_t$ is in a finite vocabulary $\mathcal{V}$ of size $N$. The probability distribution is factorized as $p_{\boldsymbol{\theta}}(\mathbf{x}) = \prod_{t=1}^{T} p_{\boldsymbol{\theta}}(x_t|\mathbf{x}_{<\mathbf{t}})$, where $\mathbf{x}_{<\mathbf{t}} = x_1 \dots x_{t-1}$. The language model is usually trained to maximize the likelihood of the true distribution $p^*(\mathbf{x})$ for any $\mathbf{x}$ from a reference dataset (the training set). In other words, given an input $\mathbf{x}_{<\mathbf{t}}$, the model learns to approximate the probability of each token from $\mathcal{V}$ being $x_t$. While this makes such a model immediately capable of scoring the probability of a given text, it also allows for the generation of new sentences. Given a conditional input (prompt) $\mathbf{z} = (z_1 \dots z_L)$, we can decode $p_{\boldsymbol{\theta}}(\mathbf{x}|\mathbf{z})$ as the continuation of $\mathbf{z}$, i.e., through the factorized representation $p_{\boldsymbol{\theta}}(\mathbf{x}|\mathbf{z}) = \prod_{t=1}^{T} p_{\boldsymbol{\theta}}(x_t|\mathbf{x}_{<\mathbf{t}}, \mathbf{z})$.

### 3.2 Reinforcement Learning for Language Models

Due to its adherence to the formal framework of Markov decision processes (Sutton & Barto, 2018), RL can be used as a solution to the generative modeling problem in the case of autoregressive tasks such as text generation (Bachman & Precup, 2015). The LLM plays the role of the agent, and each generated token represents an action $a_t$. The current version of the generated output $\mathbf{x_t}$ is part of the state $\mathbf{s_t}$ (potentially with additional information such as initial prompts). Finally, the reward $r_{t+1}$ measures the "quality" of the current output. A common strategy is to assign a zero reward for each $\mathbf{x}_t, t \neq T$ and a sentence-based reward when the final output is generated. Within this framework, any policy-based method can be employed to train or fine-tune the LLM to optimize a given objective. Indeed, RL facilitates the use of non-differentiable reward functions, enabling the optimization of test-time metrics, domain-specific targets, and human preferences (Franceschelli & Musolesi, 2024b).

A widely used RL algorithm for fine-tuning LLMs is Proximal Policy Optimization (PPO) (Schulman et al., 2017), which aims to maximize the following objective:

$$\mathcal{J}(\boldsymbol{\theta}) = \hat{\mathbb{E}}_t \left[ \min(r_t(\boldsymbol{\theta})\hat{A}_t, \texttt{clip}(r_t(\boldsymbol{\theta}), 1-\epsilon, 1+\epsilon)\hat{A}_t) \right] \tag{1}$$

where $r_t(\boldsymbol{\theta}) = \frac{\pi_{\boldsymbol{\theta}}(\mathbf{x}_t|\mathbf{z},\mathbf{x}_{<t})}{\pi_{\boldsymbol{\theta}_{old}}(\mathbf{x}_t|\mathbf{z},\mathbf{x}_{<t})}$ with $\pi_{\boldsymbol{\theta}}$ and $\pi_{\boldsymbol{\theta}_{old}}$ denoting the current and old policy models, repectively; $\mathbf{x}$ is the output sampled from the old policy given the prompt $\mathbf{z}$; and $\epsilon$ is a clipping factor used to stabilize training. The advantage $\hat{A}_t$ is usually computed through Generalized Advantage Estimation (Schulman et al., 2016) based on the full rewards $R(\mathbf{z}, \mathbf{x}) = r(\mathbf{z}, \mathbf{x}) - \beta \log \frac{\pi_{\boldsymbol{\theta}}(\mathbf{x}|\mathbf{z})}{\pi_{ref}(\mathbf{x}|\mathbf{z})}$, thus integrating a KL penalty with respect to a reference model (usually, the same model before fine-tuning), and a learned value function $v_{\phi}(\mathbf{s})$. However, learning such a value function is computationally intensive, and its training is complicated by the fact that only the last state is scored by the reward function. Moreover, the inclusion of the KL penalty as an auxiliary reward term adds

complexity to the advantage estimation process. To address these issues, Group Relative Policy Optimization (GRPO) (Shao et al., 2024) has been introduced. GRPO directly adds a KL divergence term $-\beta D_{KL}(\pi_{\boldsymbol{\theta}}||\pi_{ref})$ to the loss rather than to the single rewards, and especially obviates the need for a value function approximator by using the average reward of multiple sampled outputs as the baseline:

$$\hat{A}_{i,t} = \frac{r_i - \texttt{mean}(\mathbf{r})}{\texttt{std}(\mathbf{r})}, \tag{2}$$

with $\mathbf{r} = \{r_1, r_2, ..., r_G\}$ as the list of rewards received by each of the $G$ outputs sampled from the same prompt.

## 4 A CONTEXT-BASED SCORE FOR VALUABLE AND ORIGINAL GENERATION

Our goal is to derive a score that is able to quantify both value and originality at the same time. As discussed in depth by Csikszentmihalyi (2014), creativity depends on the context in which the product is created, as the context provides the task identification and the domain information necessary to generate and validate the outcome. In turn, the output aims to solve the given task and provide a meaningful, original contribution to the current domain. Thus, our proposed score has its roots in mutual information, which represents a quantitative way to study the relationship between contextual, prior information and a produced posterior outcome. More specifically, we start from the (point-wise) mutual information between two variables $x$ and $y$:

$$I(x, y) = h(x) - h(x|y) = h(y) - h(y|x) \tag{3}$$

where the self-information is $h(a) = -\log p(a)$, therefore:

$$I(x, y) = \log p(x|y) - \log p(x) = \log p(y|x) - \log p(y). \tag{4}$$

Let us now assume $x$ to be our input vector $\mathbf{x}$ and $y$ our output vector $\mathbf{y}$, obtaining:

$$I(\mathbf{x}, \mathbf{y}) = \log p(\mathbf{y}|\mathbf{x}) - \log p(\mathbf{y}). \tag{5}$$

We can generalize $I(\mathbf{x}, \mathbf{y})$ with two scaling factors:

$$I(\mathbf{x}, \mathbf{y}, \lambda_1, \lambda_2) = \lambda_1 \log p(\mathbf{y}|\mathbf{x}) - \lambda_2 \log p(\mathbf{y}), \tag{6}$$

where $I(\mathbf{x}, \mathbf{y})$ is just $I(\mathbf{x}, \mathbf{y}, 1, 1)$. Computing the *absolute* probability $p(\mathbf{y})$ can be difficult, as usually generative models are developed to assign *conditional* probabilities. By applying the Bayes theorem, i.e., $\log p(a|b) = \log p(b|a) + \log p(a) - \log p(b)$, we can substitute the $\log p(\mathbf{y})$ term as follows:

$$\begin{aligned} I(\mathbf{x}, \mathbf{y}, \lambda_1, \lambda_2) =& \lambda_1 \log p(\mathbf{y}|\mathbf{x}) - \lambda_2 \log p(\mathbf{y}|\mathbf{x}) - \lambda_2 \log p(\mathbf{x}) + \lambda_2 \log p(\mathbf{x}|\mathbf{y}) \\ =& (\lambda_1 - \lambda_2) \log p(\mathbf{y}|\mathbf{x}) + \lambda_2 \log p(\mathbf{x}|\mathbf{y}) - \lambda_2 \log p(\mathbf{x}). \end{aligned} \tag{7}$$

Since our goal is to find the optimal $\mathbf{y}$ for a given $\mathbf{x}$, the last term can be ignored. Moreover, we now define $\lambda_v = \lambda_2$ and $\lambda_o = \lambda_2 - \lambda_1$, thus obtaining the following objective:

$$\overline{\mathbf{y}} = \underset{\mathbf{y}}{\operatorname{argmax}}(\lambda_v \log p(\mathbf{x}|\mathbf{y}) - \lambda_o \log p(\mathbf{y}|\mathbf{x})). \tag{8}$$

Let us now consider the case where $\lambda_v, \lambda_o > 0$, for example, $\lambda_v = \lambda_o = 1$. Solving this maximization problem involves finding the target $\mathbf{y}$ that maximizes the posterior probability of $\mathbf{x}$ while also being unlikely given $\mathbf{x}$. In other words, the optimal $\mathbf{y}^*$ must be unexpected and diverse from $p(\mathbf{y}|\mathbf{x})$, but it must also be explainable by $\mathbf{x}$. $-\log p(\mathbf{y}|\mathbf{x})$, commonly known as surprisal (Tribus, 1961), is widely used to measure diversity and surprise (Barto et al., 2013), and adheres to the first requirement from the standard definition of creativity by Runco & Jaeger (2012), i.e., *originality*. Conversely, $\log p(\mathbf{x}|\mathbf{y})$ can be used to measure *value* or effectiveness, the second requirement of the definition. If the request (e.g., a problem or task) can be inferred from the outcome, the latter constitutes an appropriate instance of that task or a correct, useful solution to that problem (e.g., if the request is for a sonnet or a sci-fi screenplay, the generated artifact is valuable if identified as a poem satisfying the metrical constraints of a sonnet or as a screenplay adhering to a sci-fi theme).

In summary, the **CoVO** (**Co**ntext-based **V**alue and **O**riginality) score for a target $\mathbf{y}$ given a source $\mathbf{x}$ on a reference probability distribution $p$ is formally defined as:

$$s_{CoVO}(\mathbf{x}, \mathbf{y}, p) = \underbrace{\lambda_v \log p(\mathbf{x}|\mathbf{y})}_{\text{Value}} \underbrace{-\lambda_o \log p(\mathbf{y}|\mathbf{x})}_{\text{Originality}} \tag{9}$$

## 5   Implementation and Optimization with Autoregressive Models

We now discuss the implementation of the CoVO score with autoregressive models. Using the notation introduced above, in the context of a $\boldsymbol{\theta}$-parameterized LLM, $p(\mathbf{y}|\mathbf{x})$ can be expressed as $\prod_{t=1}^{T} p_{\boldsymbol{\theta}}(y_t|\mathbf{y}_{<t}, \mathbf{x})$. However, considering just the product of all the conditioned probabilities for an optimization problem would lead to preferring shorter sequences. To avoid this, we propose to use the $T$-th root: $\sqrt[T]{\prod_{t=1}^{T} p_{\boldsymbol{\theta}}(y_t|\mathbf{y}_{<t}, \mathbf{x})}$. By leveraging the properties of the logarithm, we obtain:

$$s_{CoVO}^{AR} = \lambda_v \frac{\sum_{i=1}^{|\mathbf{x}|} \log p_{\boldsymbol{\theta}}(x_i|\mathbf{x}_{<i}, \mathbf{y})}{|\mathbf{x}|} - \lambda_o \frac{\sum_{j=1}^{|\mathbf{y}|} \log p_{\boldsymbol{\theta}}(y_j|\mathbf{y}_{<j}, \mathbf{x})}{|\mathbf{y}|}. \tag{10}$$

It is worth noting that the vocabulary of an LLM can be extremely large, which can cause $p_{\boldsymbol{\theta}}(a|b)$ to be small even when $a$ is the most probable event given $b$. In particular, when an LLM generates $\mathbf{y}$ given $\mathbf{x}$ and then evaluates both $p_{\boldsymbol{\theta}}(\mathbf{y}|\mathbf{x})$ and $p_{\boldsymbol{\theta}}(\mathbf{x}|\mathbf{y})$, this can lead to a significant discrepancy between the magnitude of value and diversity. Since $\mathbf{y}$ has been sampled from a model based on $p_{\boldsymbol{\theta}}$, its probability would be high by definition. However, there may be various ways (possibly through synonyms) to define $\mathbf{y}$, leading to a smaller probability of $\mathbf{x}$.

Inspired by Macedo et al. (2004), we propose to counteract this problem by normalizing $p_{\boldsymbol{\theta}}(a|b)$ via $n' = \frac{n - n_{min}}{n_{max} - n_{min}}$. For probabilities, $n_{min} = 0$, while $n_{max} = \max_{v \in \mathcal{V}} p_{\boldsymbol{\theta}}(b)$, thus obtaining the overall mapping for $p_{\boldsymbol{\theta}}$: $\frac{p_{\boldsymbol{\theta}}(y_t|\mathbf{y}_{<t}, \mathbf{x})}{\max_{v \in \mathcal{V}} p_{\boldsymbol{\theta}}(\mathbf{y}_{<t}, \mathbf{x})}$. Once again, by applying the properties of logarithms, we obtain:

$$s_{CoVO}^{AR_{norm}}(\mathbf{x}, \mathbf{y}, p_{\boldsymbol{\theta}}) = \lambda_v s_v(\mathbf{x}, \mathbf{y}, p_{\boldsymbol{\theta}}) + \lambda_o s_o(\mathbf{x}, \mathbf{y}, p_{\boldsymbol{\theta}}) =$$
$$\lambda_v \frac{\sum_{i=1}^{|\mathbf{x}|} (\log p_{\boldsymbol{\theta}}(x_i|\mathbf{x}_{<i}, \mathbf{y}) - \max_{v \in \mathcal{V}} \log p_{\boldsymbol{\theta}}(\mathbf{x}_{<i}, \mathbf{y}))}{|\mathbf{x}|} -$$
$$\lambda_o \frac{\sum_{j=1}^{|\mathbf{y}|} (\log p_{\boldsymbol{\theta}}(y_j|\mathbf{y}_{<j}, \mathbf{x}) - \max_{v \in \mathcal{V}} \log p_{\boldsymbol{\theta}}(\mathbf{y}_{<j}, \mathbf{x}))}{|\mathbf{y}|}. \tag{11}$$

Calculating $p_{\boldsymbol{\theta}}(\mathbf{x}|\mathbf{y})$ is not trivial. Since LLMs are trained to complete text sequences, it is unlikely that they would generate the source text immediately after the target text (which, we should remember, is generated immediately after the source text). To address this, we consider an approximation $p_{\boldsymbol{\theta}}(\mathbf{x}|\mathbf{y}')$, where $\mathbf{y}' = \mathbf{y} + \mathbf{q}$. Here, $\mathbf{q}$ represents an additional question, such as "How would you describe this text?" or a similar formulation designed solely to increase the likelihood of generating the source text $\mathbf{x}$ (as well as alternative sources).

Once the CoVO score has been defined, its adoption in an RL framework is straightforward. As previously introduced, we can directly utilize our CoVO score as the final reward for the generated sequence. Then, the model can be trained with any policy gradient method. Our experiments leverage GRPO (Shao et al., 2024), which is a state-of-the-art choice for training language models. As introduced above, GRPO adds a per-token KL divergence term to the loss rather than to the single rewards. Usually, the KL divergence is approximated with the following estimator (Schulman, 2020): $r_{ref}(\boldsymbol{\theta}) - \log r_{ref}(\boldsymbol{\theta}) - 1$, where $r_{ref}(\boldsymbol{\theta}) = \frac{\pi_{ref}}{\pi_{\boldsymbol{\theta}}}$ and $\pi_{ref}$ is the reference policy, i.e., the model before GRPO training. In particular, GRPO aims to *minimize* the KL divergence, thus the second term $-\log r_{ref} = -\log \pi_{ref} + \log \pi_{\boldsymbol{\theta}}$ can be seen as made of two optimization problems: the minimization of the originality component $-\log \pi_{ref}$ from Equation 9, and the maximization of surprisal, or self-information, $-\log \pi_{\boldsymbol{\theta}}$ under the current model. In other words, this second term somehow trades off a portion of the originality component under the reference model (proportional to the $\beta \ll 1$ coefficient) with the surprisal under the current model. However, the first term of the KL approximation keeps the two policies closer, preventing the trained one from deviating too much and potentially disrupting the impact of the originality component in favor of the value component.

## 6   Experiments

We evaluate the effectiveness of our RL strategy through three case studies: poetry generation, mathematical problem resolution, and the tasks included in NoveltyBench[1]. In all experiments, we

---

[1]The code and results of the experiments can be found at
https://anonymous.4open.science/r/CoVO-grpo/

employ two settings, i.e., GRPO to maximize the score from Equation 11 *without* the KL divergence loss, i.e., with $\beta = 0.0$ (CoVO); and GRPO to maximize the score from Equation 11 *with* the KL divergence loss, i.e., with $\beta = 0.05$ (CoVO + KL). Both methods assume $\lambda_v = \lambda_o = 1.0$. While it is common to induce diversity at the sampling level (e.g., through min-$p$ (Minh et al., 2025) or conformative decoding (Peeperkorn et al., 2025)), we restrict our baselines to the original model and, if available, a model tuned solely on the environmental reward. Our approach is orthogonal to the chosen sampling strategy and can benefit from more advanced methods. In our evaluation, we aim at verifying whether our reward scheme can increase value and originality, regardless of how the output is sampled.

## 6.1 POETRY GENERATION

| Method | In-distribution | | | | Out-of-distribution | | | |
|---|---|---|---|---|---|---|---|---|
| | Corr. ↑ | Metric (L/S) ↑ | T-LCS ↓ | Tone ↑ | Corr. ↑ | Metric (L/S) ↑ | T-LCS ↓ | Tone ↑ |
| Meta-Llama-3-8B | 0.987 | 0.300 / 0.177 | 6.067 / 19 | $0.664_{\pm 0.069}$ | 1.000 | 0.444 / 0.132 | 5.853 / 68 | $0.621_{\pm 0.063}$ |
| + CoVO | 1.000 | 0.200 / 0.084 | 4.880 / 7 | $0.749_{\pm 0.055}$ | 0.960 | 0.267 / 0.046 | 4.933 / 6 | $0.661_{\pm 0.060}$ |
| + CoVO + KL | 0.987 | 0.333 / 0.066 | 5.107 / 9 | $0.745_{\pm 0.057}$ | 0.973 | 0.289 / 0.117 | 5.160 / 22 | $0.682_{\pm 0.059}$ |

Table 1: Aggregate results of generated poems considering both training prompts (left) and testing prompts (right). Scores on the poetical metrics are reported at the line level (L) and syllable level (S) and only consider requests for styles with specific metrical properties. Under T-LCS, we report both the mean and the maximum longest common substring across all generated poems. The mean and the 95% confidence interval are reported for tone adherence.

**Experimental Setup.** The first set of experiments concerns a very common creative task, aiming to teach the LLM to generate poems that are both more original and valuable. More specifically, we follow the approach outlined by Bradley et al. (2024) and instruct the model to write a poem in a particular style and tone. We consider the Meta-Llama-3-8B model (Grattafiori et al., 2024) as our pre-trained agent. Since we do not use the instruction-tuned model, we prompt it with some few-shot examples of the task to make it more likely to produce the desired output in the desired form (the full prompt is reported in Appendix A, together with the full training parameters). Instead of fine-tuning the entire network, we consider Low-Rank Adaptation (LoRA) (Hu et al., 2022). The original model is also used to compute the score. All sampling happens with top-$k$ ($k = 50$) and at a temperature of $1.0$. We perform a quantitative evaluation where we com-

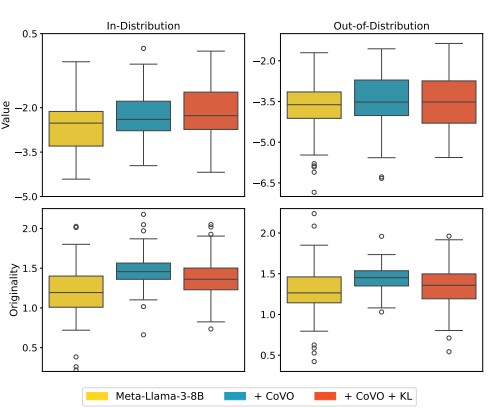

Figure 2: The distribution of value and originality (according to our scores) for the in-distribution and out-of-distribution poems generated by the baseline and our two methods.

pute poetical metrics for quality (lexical correctness of poems, adherence to line- and syllable-level constraints, and tone adherence to the request through zero-shot classification (Yin et al., 2019) with bart-large-mnli model (Lewis et al., 2020)) and for originality (accidental reproduction of existing poems). For the latter, we define a Token-based Longest Common Substring (T-LCS) score, and we use it by comparing generated poems with a reference dataset of approx. 84k public-domain poems extracted from Project Gutenberg (please refer to Appendix B for a first presentation of our GutenVerse dataset). While a generated poem can be an accidental reproduction of a protected work or a different kind of text (e.g., a song), we believe it can provide a useful evaluation tool to understand the general degree of originality.

**Experimental Results.** Table 1 reports the scores about the compliance of poetical constraints at the syllable and line levels, lexical correctness (as the ratio of poems not containing noisy text), tone adherence (as the zero-shot classification of that poem being of that tone rather than its opposite), and accidental reproduction rate (as the mean and maximum token-based longest common substring).

Overall, our CoVO-based fine-tuning leads to a higher tone adherence and lower reproduction rate, at the potential cost of metric adherence, especially without the KL loss. Indeed, its role seems to foster quality (especially in terms of metrical correctness), trading off some originality. On the contrary, not using the KL loss arguably avoids any significant reproduction, as demonstrated by the very low maximum token-based longest common substring.

Interestingly, these considerations align well with our CoVO score. Figure 2 reports the value and originality according to Equation 11 under the pre-trained model. While the two methods do not significantly differ from the baseline (which is possibly due to the opposite forces of value and originality (Varshney, 2019)), we again see that the presence of KL leads to slightly higher value, while its absence leads to slightly higher originality. However, aggregated scores, such as those presented here, might be insufficient. For a more complete overview, we also conducted a fine-grained analysis of the generated poems in Appendix C.

## 6.2 MATH PROBLEM RESOLUTION

**Experimental Setup.** The second set of experiments concerns a more practical and quantitative task, as it aims to teach the LLM to solve mathematical problems through more diverse procedures. In particular, we focus on the Mistral-based (Jiang et al., 2023) `MetaMath-Mistral-7B` model, i.e., fine-tuned with self-supervised learning on the MetaMathQA (Yu et al., 2024). It is a dataset of textual math questions paired with responses where the numerical answer is easily separable from the textual procedure. While the entire set contains 395k entries, making an additional training epoch too expensive, MetaMathQA is composed of entries from two different training sets, then augmented with various techniques: GSM8K (Cobbe et al., 2021) and MATH (Hendrycks et al., 2021). Since we are only interested in the questions, we limit our training to a single epoch over those datasets. Moreover, we exclude all questions with a tokenized length of either question or answer greater than 512, obtaining 14876 out of 14973 total entries. We separate the procedure and the answer from each solution to train our model and use the numerical answer to check the correctness of the predicted solution. The RL problem can then be formulated considering up to two rewards: our CoVO score computed on the procedure and a verifiable, extrinsic reward based on the correctness of the answer. Instead of fine-tuning the entire model, we adopt a more parameter-efficient strategy with LoRA and use the original model to perform the CoVO score computation. Following Yu et al. (2024), the outputs are obtained with a greedy strategy.

The evaluation considers both GSM8K and MATH test sets (limited to the entries with a tokenized length of question and answer smaller than 512, i.e., all 1319 entries for GSM8K and 4546 out of 5000 for MATH). We compute the percentage of correct solutions together with two diversity metrics: expectation-adjusted distinct N-grams (EAD) (Liu et al., 2022) and sentence embedding cosine similarity (SBERT) (Hong et al., 2024), which should measure syntactical and semantical diversity, respectively (Kirk et al., 2024). EAD counts the number of distinct N-grams (averaging over $N = 1 \ldots 5$) across all generated responses and removes the bias toward shorter inputs by scaling the number of distinct tokens based on their expectations. The SBERT metric computes the average of the cosine similarity between the embeddings of any possible pairs of outputs and returns 1 minus the similarity. This was originally based on Sentence-BERT (Reimers & Gurevych, 2019), we employ instead the more recent `all-mpnet-base-v2`, as suggested by their developers[2].

Following Kirk et al. (2024), we compute *cross-input* EAD and SBERT, i.e., we derive them by considering all outputs produced for a specific seed together. In addition, we also calculate *against-pretrained* EAD and SBERT. Given each input, we compare the output with the one from the pre-trained model by calculating the average expectation-adjusted distinct N-grams not present in the pre-trained model response, and 1 minus the cosine similarity between the two outputs, respectively.

**Experimental Results.** Tables 2 and 3 report the results for the GSM8K and MATH test sets. For the GSM8K test set, while all methods achieve similar results, using the CoVO score only (with and without the KL loss) leads to greater EAD diversity and to diverge more from the original model, while the presence of the math reward leads to greater accuracy, especially without KL.

The results for the MATH test confirm that the most accurate method is the one trained to optimize the CoVO score and the extrinsic reward, with a negligible trade-off in terms of diversity, since

---

[2]https://huggingface.co/sentence-transformers/bert-large-nli-stsb-mean-tokens

| Method | Accuracy ↑ | Cross-Input Diversity | | Against-Pretrained Diversity | |
|---|---|---|---|---|---|
| | | EAD ↑ | SBERT ↑ | EAD ↑ | SBERT ↑ |
| MetaMath-Mistral-7B | 77.96%(3) | 2.0071 | 0.6402 | - | - |
| + Ext | 78.18%(4) | 2.0045 | **0.6404** | $0.0081_{\pm.0021}$ | $0.0008_{\pm.0002}$ |
| + Ext + KL | 78.08%(5) | 2.0077 | 0.6401 | $0.0096_{\pm.0024}$ | $0.0011_{\pm.0004}$ |
| + CoVO | 78.12%(3) | **2.0509** | 0.6403 | $0.0854_{\pm.0062}$ | $0.0118_{\pm.0012}$ |
| + CoVO + KL | 78.12%(3) | 2.0464 | 0.6402 | $\mathbf{0.0879_{\pm.0063}}$ | $\mathbf{0.0122_{\pm.0013}}$ |
| + CoVO + Ext | **78.33**%(4) | 2.0340 | **0.6404** | $0.0628_{\pm.0056}$ | $0.0088_{\pm.0011}$ |
| + CoVO + Ext + KL | 77.95%(4) | 2.0367 | 0.6402 | $0.0638_{\pm.0057}$ | $0.0089_{\pm.0011}$ |

Table 2: Accuracy and diversity of results for the GSM8k test set. In brackets, the number of responses that exceeded the fixed maximum token limit. The best scores are highlighted in **bold**, while the worst scores are indicated with underlining. The mean and the 95% confidence interval are reported for against-pretrained diversity.

| Method | Accuracy ↑ | Cross-Input Diversity | | Against-Pretrained Diversity | |
|---|---|---|---|---|---|
| | | EAD ↑ | SBERT ↑ | EAD ↑ | SBERT ↑ |
| MetaMath-Mistral-7B | 33.55%(483) | 5.7239 | 0.8032 | - | - |
| + Ext | 33.19%(469) | 5.7187 | 0.8027 | $0.0333_{\pm.0029}$ | $0.0074_{\pm.0008}$ |
| + Ext + KL | 33.56%(476) | 5.7517 | 0.8028 | $0.0345_{\pm.0030}$ | $0.0075_{\pm.0008}$ |
| + CoVO | 33.29%(521) | 5.8219 | 0.8029 | $\mathbf{0.1457_{\pm.0049}}$ | $\mathbf{0.0339_{\pm.0016}}$ |
| + CoVO + KL | 32.79%(533) | **5.8442** | 0.8030 | $0.1479_{\pm.0050}$ | $0.0343_{\pm.0016}$ |
| + CoVO + Ext | **33.76**%(503) | 5.8114 | 0.8030 | $0.1136_{\pm.0047}$ | $0.0259_{\pm.0015}$ |
| + CoVO + Ext + KL | 33.74%(497) | 5.8082 | 0.8031 | $0.1102_{\pm.0047}$ | $0.0250_{\pm.0015}$ |

Table 3: Accuracy and diversity of results for the MATH test set. In brackets, the number of responses that exceeded the fixed maximum token limit. The best scores are highlighted in **bold**, while the worst scores are indicated with underlining. The mean and the 95% confidence interval are reported for against-pretrained diversity.

the cross-input EAD and the against-pretrained scores are still substantially higher than those from the baselines. However, removing the extrinsic reward likely pushes the model too far from its pre-trained version, causing the accuracy to decrease.

## 6.3 NOVELTYBENCH

**Experimental Setup.** Finally, we also experiment with NoveltyBench (Zhang et al., 2025), a very recent benchmark that aims to evaluate the ability of language models to produce multiple distinct and high-quality outputs. NoveltyBench contains two sets of prompts thought for eliciting diverse responses: the 'curated' partition, with 100 prompts manually curated by the paper's authors, and the 'wildchat' partition, with 1000 prompts sourced from the WildChat-1M dataset (Zhao et al., 2024). The benchmark requires a language model to generate 10 outputs per prompt using a temperature setting of 1.0, after which it computes scores for novelty and quality. To assess novelty, the 10 outputs are grouped into equivalence classes using a fine-tuned DeBERTa model (He et al., 2021), and the number of distinct classes is reported as the novelty score. To assess quality, it computes the cumulative utility of the 10 outputs, where the utility is zero if the $i$-th output has the same equivalence class as a prece-

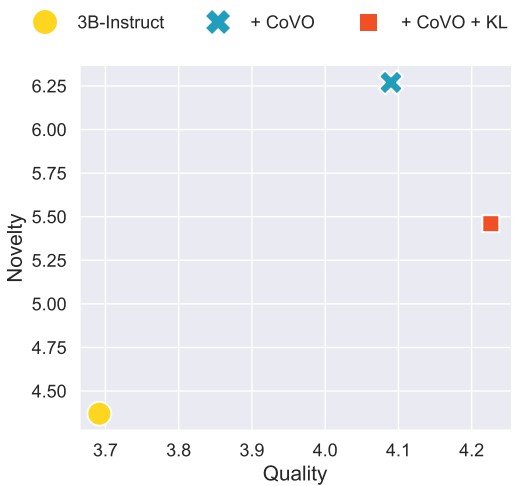

Figure 3: The mean utility (quality) and mean distinct (novelty) scores on NoveltyBench ('curated' partition) for the original model and our methods (tuned on the 'wildchat' partition).

dent output, and a calibrated reward from `Skywork-Reward-Gemma-2-27B-v0.2` (Liu et al., 2024) otherwise (we refer to Zhang et al. (2025) for full details). To evaluate our approach with NoveltyBench, we fine-tune the `Llama-3.2-3B-Instruct` model (Grattafiori et al., 2024) for a single epoch on the 'wildchat' partition (the full parameters are reported in Appendix A); then, we compute the novelty and quality scores on the 'curated' partition.

**Experimental Results.** Figure 3 reports the quality and novelty scores achieved by our methods, compared to those from the original instructed model. Optimizing for the CoVO score results in substantial improvements in both novelty and quality metrics, with the greatest gains in novelty. Moreover, these results underscore the interplay between KL loss and our CoVO score: incorporating the KL penalty tends to improve quality, but at the cost of reduced novelty.

## 7 Discussion

The CoVO score captures properties that are functionally aligned with creativity-relevant aspects of language generation, as demonstrated through both conceptual analysis and empirical results. When used as a reward function in Group Relative Policy Optimization (GRPO), CoVO drives improvements in the novelty and quality of model outputs, even with a relatively small number of optimization steps. Although KL-divergence regularization is typically employed to constrain policy shifts and preserve alignment with the base model distribution, CoVO can contribute independently to several desirable behaviors: reducing the risk of inadvertent memorization of copyrighted material, promoting diversity in generated outputs, and mitigating undesirable inductive biases introduced during pretraining.

There are a few limitations worth noting. Firstly, our score represents only a quantifiable approximation of a particular theoretical perspective on creativity, grounded in the dimensions of value and originality. For example, value has been considered from the perspective of effectiveness, while other dimensions have been proposed as well (e.g., interestingness (Boden, 1994) or monetary worth (Lepak et al., 2007)); and other classic definitions of creativity, such as that presented in Boden (2003), add a third requirement by splitting originality into novelty and surprise. Moreover, our score reflects a specific view of the evaluation of creativity based on the generated outputs and does not account for potential alternative theories (for example, arising from different cultures (Lubart, 1999)) and perspectives (Rhodes, 1961). Finally, our experiments are currently limited to only three relatively short-form text generation tasks. While their generalizability is supported by the theoretical framework discussed above, the resulting performance was experimentally evaluated for a finite number of scenarios.

## 8 Conclusion

In this paper, we presented CoVO, a novel score that quantifies the value and originality of neurally-generated text. The definition of CoVO is based on the analysis of mutual information between the model's outputs and inputs, and vice versa. We also proposed an optimization problem where a generative model aims to maximize this score to generate more creative products, and detailed how to use it in language modeling. We conducted experiments on poetry generation, math problem solving, and tasks included in NoveltyBench, exploring trade-offs in accuracy vs diversity. Effectively balancing value and originality maximization remains an open question, but our score seems to relate to domain-specific measures appropriately. In addition, fine-tuning to maximize it improves quality- and diversity-related metrics.

Our research agenda aims to extend our method to other models and tasks, to include inference-level strategies such as creativity-oriented sampling schemes, and to explore its use for evaluation (e.g., in a Best-of-N setting (Stiennon et al., 2020)) rather than solely for optimization. We also plan to investigate the definition of additional scores for capturing other potentially relevant aspects of the creative process. Despite being costly and inherently constrained (Davis, 2024), assessing whether our creativity score aligns with human judgment is another key direction for future work.

ETHICS STATEMENT

The authors are aware of the potential impact that generative technologies might have on the production of artistic outputs and, as a consequence, on human artists. This work may contribute to enhancing the quality of generated outputs. However, the authors argue that typical traits of human creativity, such as the active participation of artists in the creative process, cannot be directly replicated by machines, as also pointed out by Runco in the updated standard definition of creativity Runco (2025). The authors refer interested readers to a previous work of theirs (*citation removed for double-blind submission*), in which these themes are discussed in detail.

REPRODUCIBILITY STATEMENT

To ensure the reproducibility of our work, we provide comprehensive details throughout the paper and its appendices. The formal definition of our reward and learning scheme is presented in Section 5. Our experimental setup is described in Section 6, with full implementation details available in Appendix A. The source code for all experiments is available as supplementary material at `https://anonymous.4open.science/r/CoVO-grpo/`.

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

## A IMPLEMENTATION DETAILS

The experiments were carried out using a Linux-based local server with two 80GB NVIDIA H100 GPUs, running Python 3.11.9. All the trainings were conducted with a random seed equal to 1 (set through the `set_seed` method from the HuggingFace `transformers` library), while poems were sampled at inference time with three different seeds (1, 42, and 121). The hyperparameters were selected to accommodate the most efficient use of the available resources, and otherwise set according to their default values from HuggingFace `transformers` and `trl` libraries. Only the learning rate has been tuned for the different tasks according to their training performances.

Table 4 reports the full training parameters for the experiments on poetry generation. The prompt for generation leverages *Nothing gold can stay* by Robert Frost, *Fame is a bee* by Emily Dickinson, and *Epitaph* by William Carlos Williams for few-shot learning:

> Write a fatalistic epigram poem of high, award winning quality.
>
> Nature's first green is gold,
> Her hardest hue to hold.
> Her early leaf's a flower;
> But only so an hour.
> Then leaf subsides to leaf.
> So Eden sank to grief,
> So dawn goes down to day.
> Nothing gold can stay.
>
>
> Write an ironic quatrain poem of high, award winning quality.
>
> Fame is a bee.
> It has a song-
> It has a sting-
> Ah, too, it has a wing.
>
>
> Write a naturalistic epitaph poem of high, award winning quality.
>
> An old willow with hollow branches
> Slowly swayed his few high fright tendrils
> And sang:
>
> Love is a young green willow
> Shimmering at the bare wood's edge.
>
>
> Write a {`tone`} {`style`} of high, award winning quality.

The training phase includes requests with tone-style pairs sampled among 'dark', 'happy', 'mysterious', 'reflective' or 'romantic' for the tone, and 'ballad', 'haiku', 'hymn', 'limerick' or 'sonnet' for the style. At inference time we also consider 'cinquain', 'couplet', 'free verse', 'ode' or 'tanka' as styles and 'cutting', 'nostalgic', 'poignant', 'solemn' or 'whimsical' as tones.

Instead, the prompt used for computing $p_{\boldsymbol{\theta}}(\mathbf{x}|\mathbf{y})$ is:

> Describe the style of the following poem in two words:
>
> {`prova`}
>
> I would describe it as a

| Parameter | Value |
|---|---|
| Total batches | 100 |
| Batch size $B$ | 4 |
| Gradient accumulation steps | 8 |
| Max new tokens | 256 |
| Temperature | 1. |
| Top-$k$ | 0 |
| Optimizer | Adam |
| Learning rate | 1e-5 |
| Max gradient normalization | 100. |
| Rank (LoRA) | 16 |
| $\alpha$ parameter (LoRA) | 32 |
| Dropout (LoRA) | 0.05 |
| Training iterations | 1 |
| Scale rewards | True |
| $\beta$ (when used) | 0.05 |
| Number of generations $G$ | 4 |

Table 4: Training parameters for poetry generation.

| Parameter | Value |
|---|---|
| Total epochs | 1 |
| Batch size $B$ | 4 |
| Gradient accumulation steps | 8 |
| Max new tokens | 512 |
| Temperature | 1. |
| Top-$k$ | 0 |
| Optimizer | Adam |
| Learning rate | 1e-6 |
| Max gradient normalization | 100. |
| Rank (LoRA) | 16 |
| $\alpha$ parameter (LoRA) | 32 |
| Dropout (LoRA) | 0.05 |
| Training iterations | 1 |
| Scale rewards | True |
| $\beta$ (when used) | 0.05 |
| Number of generations $G$ | 4 |
| Reward for correct answer | +1. |

Table 5: Training parameters for math problem solving.

Finally, the zero-shot classification for the tone adherence is performed with the given tone and 'not' plus the given tone as the candidate labels (e.g., if the required tone is 'happy', the two labels are ['happy', 'not happy']).

On the contrary, Table 5 reports the full training parameters for math problem resolution. We also adopted the same two different prompts from Yu et al. (2024), i.e.:

Below is an instruction that describes a task. Write a response that appropriately completes the request.

### Instruction:
{question}

### Response:

at training time and

| Parameter | Value |
|---|---|
| Total epochs | 1 |
| Batch size $B$ | 4 |
| Gradient accumulation steps | 8 |
| Max new tokens | 512 |
| Temperature | 1. |
| Top-$k$ | 0 |
| Optimizer | Adam |
| Learning rate | 1e-4 |
| Max gradient normalization | 100. |
| Rank (LoRA) | 16 |
| $\alpha$ parameter (LoRA) | 32 |
| Dropout (LoRA) | 0.05 |
| Training iterations | 1 |
| Scale rewards | True |
| $\beta$ (when used) | 0.05 |
| Number of generations $G$ | 4 |

Table 6: Training parameters for NoveltyBench.

> Below is an instruction that describes a task. Write a response that appropriately completes the request.
>
> ### Instruction:
> {question}
>
> ### Response: Let's think step by step.

at inference time. Instead, for computing $p_{\boldsymbol{\theta}}(\mathbf{x}|\mathbf{y})$ we used the following:

> Below is a response that appropriately completes a request. Write the instruction that describes the task.
>
> ### Response:
> {response}
>
> ### Instruction:

Finally, Table 6 reports the full training parameters for NoveltyBench. At training and inference time, we simply adopt the following prompt:

```
user

{prompt}
assistant
```

where `user` and `assistant` are keywords used by the model to identify different roles in the chat. Instead, for computing $p_{\boldsymbol{\theta}}(\mathbf{x}|\mathbf{y})$ we used the following:

> Below is a response that appropriately solves a task. Write the instruction that describes the task.
>
> ### Response:
> {`response`}
>
> ### Instruction:

## B  GUTENVERSE DATASET

To evaluate the accidental reproduction rate of generated poems, we introduce the GutenVerse dataset, which comprises over 84,000 public-domain, English-written poems extracted from Project Gutenberg. While generated poems can reproduce different content, e.g., songs or copyrighted material, we believe this can provide a useful indication of how likely a text is to be original or not.

To derive our dataset, we started from *Gutenberg, dammit*[3], a corpus of every plaintext file in Project Gutenberg (up until June 2016). We selected all the text files whose metadata report English as the language, public domain as copyright status, *poetry* among the subjects or *poems* or *poetical work* in the title, and that were not a translation of another book. Then, we applied a series of rules (e.g., about the verse length) to automatically extract the titles and poems from all the selected text files, and we defined our GutenVerse dataset. While it can still contain content that is not poetry (e.g., a table of contents formatted very uncommonly), the poems can be effectively used to measure the overlap between real and generated text.

We also released a datasheet (Gebru et al., 2018) for the GutenVerse dataset that can be found, together with the code used to create it, at: `https://anonymous.4open.science/r/GutenVerse-DD32/`.

## C  DETAILED ANALYSIS OF THE GENERATED POEMS

We now present some noteworthy generated poems to provide a detailed qualitative discussion of our methods and our score.

The baseline model `Meta-Llama-3-8B`, while producing prosaic text (without line breaks) occasionally, is the most conservative method. This results in its poems being usually well-formatted but also somehow *banal*, and more prone to reproducing existing works. For example, when asked to write a dark hymn, it produced the very famous two lines "i am the master of my fate: / i am the captain of my soul". In terms of the CoVO score, it is noteworthy that the poem with the lowest originality component achieves the highest T-LCS score. This strongly suggests that the model has memorized the hymn *All Things Bright and Beautiful* by Cecil Frances Alexander and reproduced it almost entirely, with only minor alterations in punctuation—enough to prevent the LCS score from being even higher. We report it in Table 7. Instead, for the overall score, the value part becomes the most relevant: the adherence to the requirement seems to be the reason behind the highest and lowest scores, as shown in Table 8.

Regarding the fine-tuned models, the one trained without the KL loss exhibits the opposite behavior. It never reproduces existing poems, but occasionally exploits the CoVO score in adversarial ways—for example, by learning to generate specific words associated with the requested tone. In one instance, it even learned to simply repeat the style-tone prompt itself, which resulted in the highest possible score, albeit at the cost of very low originality. Instead, maximizing the originality component makes the model produce noteworthy outputs, but without satisfying stylistic constraints, as reported in Table 9. In general, optimizing the CoVO score without the KL loss leads the model to deviate from memorized poems but also from metrical rules, and leads it to focus on the tone (both the explicit, requested tone and that intrinsic to the poetic style). For example, its limericks maintain the typical playful tone, but only once adhere to the traditional "There [once] was a man" starting line and only partially, as it starts with "There was an old dog". Interestingly, it is still not

---

[3]`https://github.com/aparrish/gutenberg-dammit/`

the highest-valued limerick; the best in terms of value component better shares the jocking nature of limericks (see Table 10).

On the contrary, including the KL loss makes the model adhere more to the required style; for instance, the majority of limericks start with different subjects of the classic "There [once] was a man". In general, the KL loss seems to help the model preserve coherence and increase the CoVO score in a more meaningful way, e.g., the repetitions are quite rare (though, when present, they lead again to the highest value scores), as well as the regurgitation of existing poems. Interestingly, this method also leads to the highest-scoring poem from an out-of-distribution tone-style pair, as reported in Table 11, together with the best poem that does not contain adversarial exploitation of the scores (e.g., repetitions or mention of the request), which, though not perfectly adhering with the syllable counts, share all the features of a proper haiku.

| Happy hymn | |
|---|---|
| all things bright and beautiful, | |
| all creatures great and small, | |
| all things wise and wonderful; | |
| the lord god made them all. | |
| each little flower that opens, | |
| each little bird that sings, | |
| he made their glowing colors, | |
| and spread their little wings. | |
| the rich man in his castle, | |
| the poor man at his gate, | |
| god made them high or lowly, | |
| and ordered their estate. | |
| the purple headed mountain, | |
| the river running by, | |
| the sunset and the morning | |
| that brightens up the sky; | |
| $V = -2.118$ | $O = 0.219$ |

Table 7: The poem generated by the pre-trained model with the highest T-LCS score and the lowest originality score.

| Romantic sonnet | |
|---|---|
| when to the bat and the moth | |
| the world becomes a moon of beauty | |
| as twilight's cloak unfurls, | |
| and dreams and dreams and more dreams | |
| fill all of this fair air with poetry, | |
| a kiss is worth more than diamonds pure | |
| or sapphires rich and glinting | |
| like eyes of lovers in romance, | |
| where love is just like dew on wings of swans; | |
| and all that is beautiful is sweet, | |
| so love me as our love is new, | |
| my dear, for i love you! | |
| $V = -0.449$ | $O = 1.568$ |
| Nostalgic ode | |
| there was an old tree growing in a wood, | |
| its fruit was red, and its leaves were blue. | |
| an ant ran up it, and then down again, | |
| it turned into a purple snail in june. | |
| $V = -6.845$ | $O = 1.206$ |

Table 8: The poems generated by the pre-trained model with the highest (top) and lowest (bottom) overall CoVO score (top).

| Mysterious haiku | |
|---|---|
| when no one sees, | |
| a child plays. | |
| i love what could be. | |
| no one believes? | |
| so they know her mind: | |
| so the bird sings | |
| for me alone | |
| this evening of dark rain? | |
| $V = -2.501$ | $O = 2.177$ |
| Whimsical couplet | |
| we need no art for that; | |
| but in wild nature see | |
| so sweet, it is a gift | |
| sang at every bird and flower. | |
| $V = -2.103$ | $O = 1.960$ |

Table 9: The poems generated by the CoVO-based fine-tuned model with the highest originality scores from training (top) and testing (bottom) tone-style pairs.

| Happy limerick | |
|---|---|
| The young man was a master of the rhyme | |
| and in love with the young lass to this time. | |
| So for her first night with her he wrote | |
| a new form to try in order the note… | |
| A true Limerick for her delight. | |
| Oh so happy and joyous the first night of love! | |
| $V = -0.818$ | $O = 1.370$ |
| Dark limerick | |
| There was an old dog from Japan | |
| Who fell in love with a harem of cans. | |
| He would hide there on hot afternoons, | |
| Till the sun turned his hair purple fawns. | |
| One day, while watching a game in blue jeans, | |
| He dropped by mistake the key to his knees. | |
| He gave the other a quick whiplash and then, | |
| Said to himself, "Ouch! my balls are in danger!" [...] | |
| $V = -1.261$ | $O = 1.354$ |

Table 10: Two limericks from the CoVO-based fine-tuned model: above, the one with the highest value score; below, the only generated one that starts with the classic first line.

| Romantic haiku | |
|---|---|
| spring winds | |
| carry in the leaves. | |
| love whispers. | |
| $V = -1.343$ | $O = 2.049$ |
| Whimsical couplet | |
| no more of one or two | |
| let there be more. | |
| one's never new | |
| but two and more's a rarity. | |
| $V = -1.361$ | $O = 1.718$ |

Table 11: Two of the highest-scoring poems from the CoVO+KL-based fine-tuned model.

