# OpenReview forum: "Thinking Outside the (Gray) Box: A Context-Based Score for Assessing Value and Originality in Neural Text Generation"
_ICLR.cc/2026/Conference — ICLR 2026 Conference Withdrawn Submission_

### Official Review · Reviewer_RULH · 2025-11-01

**Soundness:** 2
**Presentation:** 3
**Contribution:** 2
**Rating:** 4
**Confidence:** 4

**Summary:**

Disclosure: Claude is used to refine the review.

This paper proposes CoVO, a context-based score derived from mutual information to assess value and originality in neural text generation. The score combines two components: value (log p(x|y), measuring how well the input can be inferred from output) and originality (-log p(y|x), measuring surprisal). CoVO is used as a reward signal in GRPO to fine-tune LLMs for creative tasks. Experiments on poetry generation, math problem solving, and NoveltyBench demonstrate that the method can improve diversity without sacrificing quality.

**Strengths:**

- The tension between quality and diversity in LLM generation is important, and balancing value with originality is a meaningful objective.
- Experiments span multiple domains. Testing on poetry, mathematics, and NoveltyBench shows effort to validate across different creative tasks.
-  The proposed method is simple. Meanwhile, the paper provides detailed implementation guidance, e.g., how to compute the score with autoregressive models and integrate it with GRPO.
- The release of GutenVerse dataset for poetry evaluation is a useful resource.

**Weaknesses:**

- My main concern is that this paper compares with no existing work on improving creativity of language models, such as DivPO, DRA-GRPO, and DARLING.
- The leap from mutual information to "creativity" lacks rigorous justification. I don't get the claim that log p(x|y) measures "value" - if y is relevant to x but y itself is ungrammatical/low-quality, won't log p(x|y) still be high?
- Computing p(x|y) for autoregressive models requires a workaround (adding prompt q to make y' = y + q) and is not guaranteed to approximate the true posterior.

**Questions:**

- Why is log p(x|y) evaluating "value"? If y is ungrammatical/low-quality but still relevant to x, would log p(x|y) still be high?

---

> ### Author Response · Authors · 2025-11-19
>
> We thank the reviewer for the helpful comments. Please find the response to the raised concerns below.
>
> Weaknesses:
>
> - We decided to limit our evaluation to the original model to better highlight the contribution of our score. While we are aware that other strategies exist, they usually focus on diversity alone, as is the case for DivPO and DRA-GRPO, and require an environmental reward to exist (which only happens in one out of three of the tested case studies). Moreover, the DARLING paper was published on arXiv less than a month before our submission, thus making it impossible to consider for comparison. However, we agree that our paper would benefit from additional comparisons with the most similar methods, and we plan to do so in a revised version of the manuscript.
>
> - Yes - we are aware that this is a potential limitation of our formulation. However, we never claimed our formula to be *universal*: as stated in the Discussion, “value has been considered from the perspective of effectiveness, while other dimensions have been proposed as well”, including correctness. We will make it clearer in the revised version of the paper.
>
> - While we agree that this workaround is not perfect, a workaround is needed anyway, since the true posterior is computationally infeasible. We acknowledge that it is not possible to theoretically guarantee that our workaround approximates the true posterior, since it is ideally possible to find a $q$ that changes the posterior distribution dramatically. However, in practice, as we merely define $q$ as a neutral question to simplify the posterior computation, we do not believe it deviates significantly from the true posterior and can be effectively used in its place.
>
> Questions:
>
> - As we described in lines 207-211 and in the Discussion, our value definition is based on the idea that a valuable artifact needs to be effective and appropriate in solving a task. As we stated before, we agree that an overarching value definition should also include a notion of correctness (as well as other properties), which is not present here. In general, it is possible to obtain a high $\log p(x|y)$ with a grammatically incorrect text; however, the use of the KL divergence penalty can prevent this (and has effectively prevented this in our experiments).

---

### Official Review · Reviewer_Gznv · 2025-11-01

**Soundness:** 2
**Presentation:** 1
**Contribution:** 1
**Rating:** 2
**Confidence:** 4

**Summary:**

The paper proposes CoVO (Context-based score for Value and Originality), an information-theoretic score designed to evaluate and encourage both value and originality in neural text generation. The authors derive CoVO from mutual information between model inputs and outputs and demonstrate how it can be used as a reward in reinforcement learning (via GRPO) to fine-tune large language models for creative tasks. Experiments are presented across three domains: poetry generation, mathematical problem solving, and the NoveltyBench benchmark. The authors claim that optimizing for CoVO enhances creativity-oriented metrics without sacrificing correctness.

**Strengths:**

- Clear motivation: addressing the diversity–quality trade-off in creative LLM generation.
- Application of CoVO within reinforcement learning is technically sound and compatible with modern LLM fine-tuning frameworks (e.g., GRPO).
- The experiments span both creative (poetry) and analytic (math reasoning) domains, suggesting some generality.

**Weaknesses:**

- Lack of comparative evaluation. The paper does not benchmark CoVO against existing novelty/diversity metrics (e.g., Diversity Is All You Need, intrinsic rewards, novelty search).
- All experiments rely on similar foundation model families (llama). Experiments on a different family of foundation models would be useful (e.g., Qwen).
- Qualitative analysis or examples demonstrating that outputs are actually more creative or original would be useful to see.
- Reported improvements are small and not statistically significant.

**Questions:**

- The CoVO objective may be gameable (Goodhart’s law). Did the authors see any reward hacking happening? How might we prevent such reward hacking of quantitative measures from happening?
- Are there possible failure modes or adversarial cases where optimizing CoVO harms creativity instead?

---

> ### Author Response · Authors · 2025-11-19
>
> We thank the reviewer for the feedback and useful comments. Please find our responses to the raised concerns below.
>
> Weaknesses:
>
> - We decided to limit our evaluation to the original model to better highlight the contribution of our score with respect to it. While we recognise the presence of other strategies in the literature, many of them emphasise diversity alone and rely on the environment to provide a quality-related reward signal, which was absent in two of the three scenarios we investigated. Other methods, including novelty search, differ conceptually from our framework and therefore fall outside the scope of direct comparison. However, we agree that our paper would benefit from additional comparisons with the most similar methods. This will be addressed in a revised version of the paper.
>
> - We agree with the reviewer on this point. Although the method should, in principle, be independent of the model family, certain models may exhibit characteristics that either favour or hinder our reward function. We plan to extend our experiments to additional model families in future work.
>
> - While we decided not to insert this in the main body due to length constraints, we discussed some qualitative findings in the Appendix. Nonetheless, we agree that this analysis could have been expanded to all three case studies.
>
> - While some improvements are small and their confidence intervals overlap, this does not hold for all experiments. For example, the scores for the NoveltyBench experiments are significantly higher, and the same applies to some metrics for poetry generation (such as T-LCS) and math problem solving (all diversity metrics, even though the improvements are small).
>
> Questions:
>
> - As we reported in Appendix C, we experienced some reward hacking when the CoVO score was used alone. However, with the KL divergence in place, we did not see any adversarial exploitation of the reward model. We are aware that this should have been discussed in the main body of our paper rather than in the Appendix, and we plan to move the discussion to the main body in the revised version of the paper.
>
> - There are potential failure modes, which may arise due to reward hacking (see above) or due to the exploitation of one of the two scores alone, e.g., only maximizing originality (outputting something completely unpredictable and thus meaningless) or only maximizing value (repeating the request, which is not original at all and helps hack the value-based reward function). However, retaining both scores and incorporating the KL divergence largely mitigated these failure modes in our experiments.

---

### Official Review · Reviewer_ZVTj · 2025-11-11

**Soundness:** 2
**Presentation:** 2
**Contribution:** 2
**Rating:** 4
**Confidence:** 3

**Summary:**

The paper proposes CoVO, a context-based score rooted in mutual information theory, to quantify value as log p(x|y) and originality as -log p(y|x). This may promote divergence from the model's distribution in LLM-generated text. It employs CoVO as a reward in GRPO fine-tuning to enhance LLM creativity. Tested on poetry generation, math problem solving, and NoveltyBench tasks, the results demonstrate improvements: poetry shows higher tone adherence and lower reproduction rates; math yields better accuracy and diversity; NoveltyBench boosts novelty and quality.

Strength:
1. The paper is well written and easy to understand.
2. Comprehensive empirical validation across varied tasks.

Weaknesses:
1. The concepts of value and originality are not theoretically defined.
2. Using a reward model as a proxy is known to induce the reward hacking problem.

Given the vagueness in the task definition, I am leaning towards rejection.

**Strengths:**

1. The paper is well written and easy to understand.
2. Comprehensive empirical validation across varied tasks.

**Weaknesses:**

1. The concepts of value and originality are not theoretically defined. The use of P(x|y) and P(y|x) is unjustified. In addition, this also leads to the second issue.
2. Using a reward model as a proxy is known to induce the reward hacking problem. The reward score can be optimized to a higher level without necessarily improving the proxied quality. Also, the metrics in evaluation (EAD, T-LCS, SBERT, etc.) might suffer from a similar problem as reward hacking (following Goodhart's law).

**Questions:**

N/A

---

> ### Author Response · Authors · 2025-11-19
>
> We thank the reviewer for the insightful comments. Please find our responses to the points raised by the reviewer below:
>
> 1) We understand that the definition of value and originality should have been made clearer. Nevertheless, we respectfully disagree with the conclusion that the two formulas lack justification. As clearly stated in lines 202-211, $-\log p(y|x)$ is a widely used measure in the context of originality and surprise, and there is plenty of evidence in the literature, e.g., [1-3]. On the other hand, while less common, $\log p(x|y)$ can arguably be mapped to the effectiveness and the appropriateness of the answer given the question, two properties central in the definition of creative value [4].
>
> 2) We are aware that the use of a reward model can lead to reward hacking; however, this is true for any reward modeling strategy, and it is not limited to our method. In other words, this appears as a concern that is orthogonal to the problem at hand. Moreover, we explicitly acknowledged this in Appendix C in our paper, where we presented certain qualitative examples of generated poems that exploit the reward model adversarially. We agree that this should have been discussed in the Discussion section as well, and we plan to do so in the revised version of the paper. Finally, we do not understand how the use of specific, quantitative metrics, extensively used by several other articles in the literature, can lead to reward hacking, as our reward model is not based on them.
>
> References:
>
> [1] R. Bunescu and O. Uduehi, “Learning to Surprise: A Composer-Audience Architecture”, in *Proc. of the 10th International Conference on Computational Creativity (ICCC’19)*, 2019.
>
> [2] K. Burns, “Computing the creativeness of amusing advertisements: A Bayesian model of Burma-Shave’s muse,” *Artificial Intelligence for Engineering Design, Analysis and Manufacturing*, vol. 29, no. 1, pp. 109–128, 2015.
>
> [3] K. Grace and M. L. Maher, “What to expect when you’re expecting: The role of unexpectedness in computationally evaluating creativity,” in *Proc. of the 5th International Conference on Computational Creativity (ICCC’14)*, 2014.
>
> [4] M. A. Runco, “Updating the Standard Definition of Creativity to Account for the Artificial Creativity of AI,” *Creativity Research Journal*, vol. 37, no. 1, pp. 1–5, 2023.

---

### Author Response · Authors · 2025-12-04
**Message to the AC**

**Message to the AC**

First of all, we would like to thank the AC for their work and time. Since the rebuttal period was cut short, there was insufficient time for reviewers to engage with our responses and clarifications. In our responses, we tried to address all the reviews very carefully. Please do not hesitate to contact us if you need any clarification or information.

The Authors

---

### Note · Authors · 2026-01-16

I have read and agree with the venue's withdrawal policy on behalf of myself and my co-authors.